# Verification of Accuracy of Unmanned Aerial Vehicle (UAV) Land Surface Temperature Images Using In-Situ Data

**Bonggeun Song [1] and Kyunghun Park [2,*]**

[1] Institute of Industrial Technology, Changwon National University, 20 Changwondaehak-ro Uichang-gu Changwon-si, Gyeongsangnam-do 641-773, Korea; pureguy55@gmail.com

[2] School of Civil, Environmental and Chemical Engineering, Changwon National University, 20 Changwondaehak-ro Uichang-gu Changwon-si, Gyeongsangnam-do 641-773, Korea; landpkh@changwon.ac.kr

[*] Correspondence: landpkh@changwon.ac.kr; Tel.: +82-10-9992-7567; Fax: +82-55-281-3011

**Abstract:** The accuracy of land surface temperatures (LSTs) acquired by an unmanned aerial vehicle (UAV) was verified by comparison with in-situ LSTs of various land cover materials at the Changwon National University Campus, Changwon City, South Korea. UAV imaging and in-situ measurements were performed on July 31st and August 2nd, 2019. During the in-situ measurements, LST was measured at 160 points using an infrared thermometer. The linear regression model between the UAV and in-situ measurements exhibited a very high correlation on both days, with $R^2$ values greater than 0.7004. The root mean square error (RMSE), however, was 4.030 °C on July 31st and 5.446 °C on August 2nd, and it also varied depending on the land cover type. These results may depend on various factors, such as the field of view and performance of the TIR (Thermal infrared radiance) camera, as well as the weather and atmospheric conditions. Accurately diagnosing the thermal characteristics of urban areas based on the spatial elements can be used to accurately analyze the thermal characteristics of urban areas and to make effective policy decisions. Techniques for verifying and improving the accuracy of UAV TIR LST data for various land cover materials are required to enable precise investigation of the thermal characteristics of urban areas.

**Keywords:** UAV, Land surface temperature; Remote Sensing; Urban heat island; Land cover

## 1. Introduction

Urban areas are subjected to thermal stresses, such as heat waves and tropical nights, due to the urban heat island (UHI) phenomenon, in which urban areas are hotter than surrounding suburban areas. This phenomenon occurs because the land surface temperature (LST) increases as urban green areas are replaced by artificial land cover materials with high solar radiation absorption, and the temperatures of urban areas become higher than those of suburban areas [1]. Also, UHI is caused by differences in evaporation and radiation absorption during the day and heat storage on the surface at night. In order to reduce the UHI, it is necessary to identify the effect of LST on the phenomenon by considering various spatial characteristics of urban areas [2–5]. In this respect, knowledge of LSTs is important for analyzing the thermal characteristics of urban areas and the UHI.

With the recent development of remote sensing technologies, many studies have been conducted to analyze LST using satellite images. Thermal infrared (TIR) satellite images can be used to monitor UHI on a large scale and analyze time-series changes because they can periodically acquire LST data over large areas [6–9]. They can also analyze LST characteristics based on their spatial patterns, and thus can be useful in identifying the distribution of the UHI resulting from urban development [10,11]. The types of TIR satellite images include the Moderate Resolution Imaging Spectra radiometer (MODIS), Landsat, and the Advanced Spaceborne Thermal Emission and Reflection

Radiometer (ASTER), each with specific spatiotemporal resolution [12]. TIR satellite images have limitations in identifying the thermal characteristics of urban areas as a function of the complex and diverse land cover materials due to low spatial resolution (MODIS: 500 m, Landsat: 120 m, ASTER: 90 m) [13–15]. Moreover, there are limitations in acquiring spatiotemporal satellite images because satellite images are captured according to a fixed time and route [1,16,17]. To overcome these issues, studies have been actively conducted using unmanned aerial vehicles (UAVs).

UAVs can capture LST data when equipped with TIR cameras (in the bandwidth 3.5-14 μm). They can precisely identify LST characteristics as a function of the land cover material because, unlike satellite images, they can acquire high-resolution images. They can also collect image data without time and location constraints [18]. While satellites operate at high altitudes and thus have problems acquiring accurate image data due to weather conditions in the atmosphere, UAVs can acquire more accurate and precise LST than satellite images because they fly at low altitudes of around 100 m and thus mitigate the influence of weather conditions.

For this reason, UAVs with TIR cameras are being used in various areas of research to observe LST patterns. Naughton and McDonald (2019) observed LST in a complex urban environment and reported that LST is affected by the characteristics of land cover materials, weather, urban geometry, and traffic [1]. Kraaijenbrink et al. (2018) performed mapping of LST on the Lirung Glacier in the central Himalayas by comparing UAV TIR LSTs, Landsat 8 TIR images, and in-situ LSTs [19]. Tucci et al. (2019) analyzed the thermal characteristics of dry-stone wall terraced vineyards in Chianti, Tuscany, Italy, and detected microclimate dynamics induced by dry-stone terracing [20]. Gaitani et al. (2017) produced a map that combined LST, albedo, and apparent thermal inertia using a UAV to improve the classification of fine land cover materials and energy balance models in urban areas and to acquire microclimatic information [18]. Kang et al. (2018) derived the usability of UAV TIR LSTs in analyzing the thermal environment of urban green areas. As described above, many studies have analyzed LST using TIR cameras mounted on UAVs [21]. However, few studies have verified the accuracy of LSTs acquired from UAV TIR cameras through in-situ measurements. Although some studies compared UAV TIR LSTs with in-situ LSTs for several land cover materials using thermal imaging cameras or contact-type surface thermometers [19,21–23], the studies that verified the accuracy of UAV TIR LSTs for various land cover materials distributed in complex urban areas are insufficient. TIR satellite images are used in urban planning to alleviate the UHI by verifying the accuracy of LSTs and correcting them using the measured data [11,24–28]. Therefore, to identify the thermal characteristics of the land surface using LST data collected from UAVs, and to utilize such data for improving the thermal environment in urban areas, it is necessary to first verify their accuracy by comparing them with measured data.

Therefore, in this study, an attempt was made to analyze the accuracy of the LSTs acquired from a UAV TIR camera of various urban land cover materials in an area of the Changwon National University Campus located in Changwon City, South Korea. To this end, high-resolution (2 cm) LST images were collected using a UAV TIR camera and in-situ measurements were performed considering various land cover types to measure LST. In addition, the UAV TIR LSTs and in-situ LSTs for the various measurement points were compared.

## 2. Materials and Methods

### 2.1. Study Area

This study was conducted in an area of the Changwon National University Campus (N35°14′30″, E128°41′50″) located in Changwon City, South Korea (Figure 1). Changwon City has mild weather with an average annual temperature of approximately 15 °C, and an average annual precipitation of 1,400 mm (https://www.changwon.go.kr). It has four distinct seasons—in summer (June to August), heat waves with temperatures higher than 30 °C and tropical nights are observed, whereas intensive rainfall occurs during the rainy season [29]. The city is located in a basin surrounded by mountains with elevations of approximately 600 m. As air circulation is not good in the city, the UHI and air quality problems occur constantly in the area.

UAV imaging and in-situ measurements were performed in a 0.15 km² area in the Changwon National University Campus, where the engineering college building is located. Five-story buildings are located around pedestrian paths in the area. The paths are approximately 20 m in width and are covered with sidewalk bricks. Wooden decks, lawns, and trees are located around the paths. The trees are approximately 4 m high and do not have large leaves. Roads paved with asphalt occur around the paths. The building roofs are covered with water-proof paint in various colors, such as gray, green, and white. The paint on some of the building roofs has been removed considerably by aging.

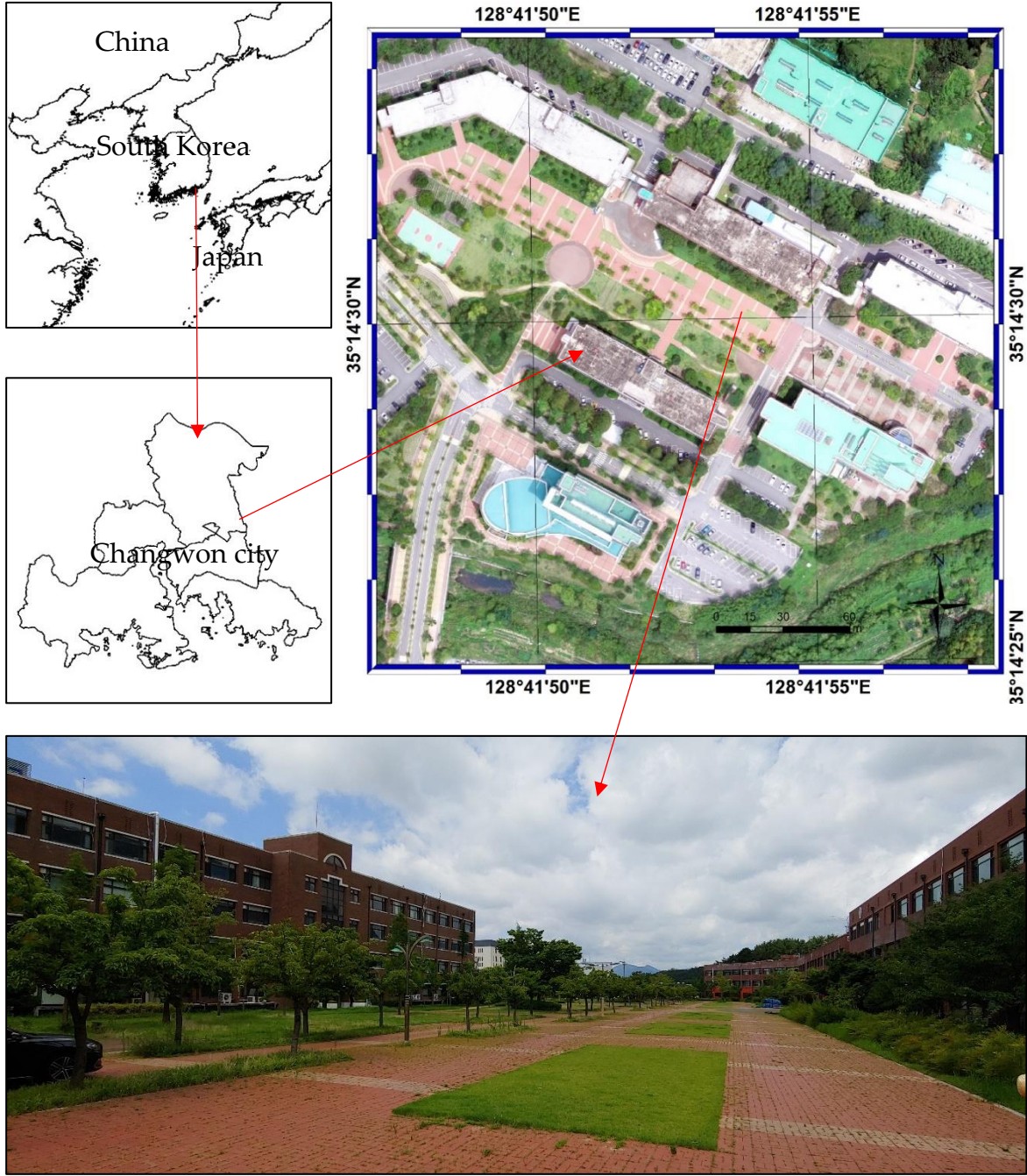

**Figure 1.** Location map and images of the study area (Changwon National University Campus).

*2.2. Acquisition of UAV TIR images*

The UAV TIR images were captured on July 31st and August 2nd, 2019 at 12 noon, when the influence of shadow was minimal due to the highest solar altitude. As for the weather conditions observed by the nearby automatic weather measurement device (http://bangjae.changwon.go.kr), the temperature was approximately 32.6 °C with no rainfall and almost no wind (wind speed: 0 m/s) on July 31st, although it was cloudy. On August 2nd, it was very hot with a temperature of 37.2 °C, but the rainfall and wind speed conditions were the same as those on July 31st—it was a clear day with few clouds.

The UAV TIR LST images were captured with a FLIR Vue Pro R TIR camera (spectral range: 7.5–13.5 μm, accuracy: ± 5 °C, and emissivity: 0.98) mounted on a DJI Inspire 1. Using the longwave radiance obtained from a FLIR Vue Pro R TIR camera, the LST is calculated using the following equation in the Pix4D Mapper program. Table 1 shows the detailed specifications of the devices used for capturing UAV images.

$$\text{UAV LST (℃)} = 0.04 \times \text{Longwave radiance} - 273.15$$

UAV operation began at 12 noon, and TIR images of the study area were captured three times for 30 minutes. The UAV flight altitude was 150 m, and the image-overlapping ratio was set at 85 %. Orthoimages were produced from the captured images using the Pix4D Mapper software, and the spatial resolution was set at 0.2 cm.

**Table 1.** Detailed specifications of Inspire 1 and FLIR Vue Pro R.

| Item | Detailed specifications |
|---|---|
|  **Inspire 1** | Weight: 2.935kg <br> Max. flight time: 18~20 min <br> Max. speed: 22m/s <br> Operating temperature: -10 to 40°C |
|  **FLIR Vue Pro R** | Size: 58 × 45 mm <br> Spectral range: 7.5–13.5 μm <br> Accuracy: ± 5 °C <br> Weight: 92–113 g <br> Operating temp. range: -55–95 ℃ <br> Field of View (FOV): 6.8 mm, 45° × 35° <br> Resolution: 336 × 256 pixels |

*2.3. In-situ measurement of LST*

A total of 160 points were selected for the in-situ measurement, taking into consideration the land cover types distributed in the study area shown in Figure 2 and the measurement time. Fewer measurement points were selected for the barren (2 points), urethane (4), concrete (4), wooden deck (6), and gravel (8) types that occupied relatively small areas. On the other hand, many points (34 each) were selected for the sidewalk brick and asphalt types, which covered the pedestrian space and roads, while 44 points were selected for the grassland type because it occupied the largest area. In addition, the green urethane (9), gray concrete (10), and white urethane (5) that covered the building roofs were also measured. Figure 3 shows the actual view of the land cover types.

　　　The in-situ measurement was performed by three teams of two persons. One person performed the measurement at a 10 cm height from the land surface using an infrared thermometer (Testo 381, accuracy: ± 1.5 °C, emissivity: 0.98), and the other recorded the measurements in a field book. At each measurement point, measurement was performed three times and the average value was calculated to determine the LST of the point.

　　　The in-situ measurement was initiated at the same time as the UAV operation. The measurement was completed within 30 minutes to minimize changes in LST with time.

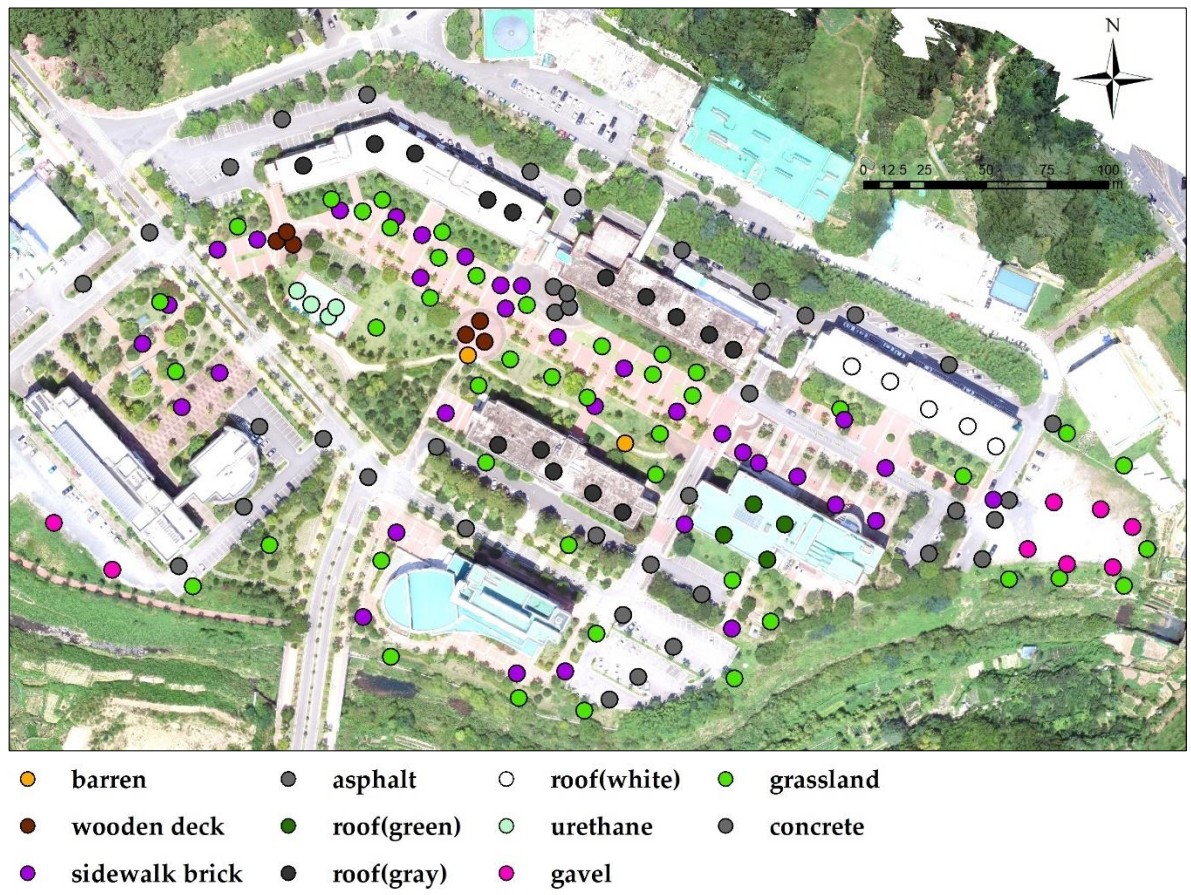

**Figure 2.** Image of measurement points, which are color-coded by land cover type.

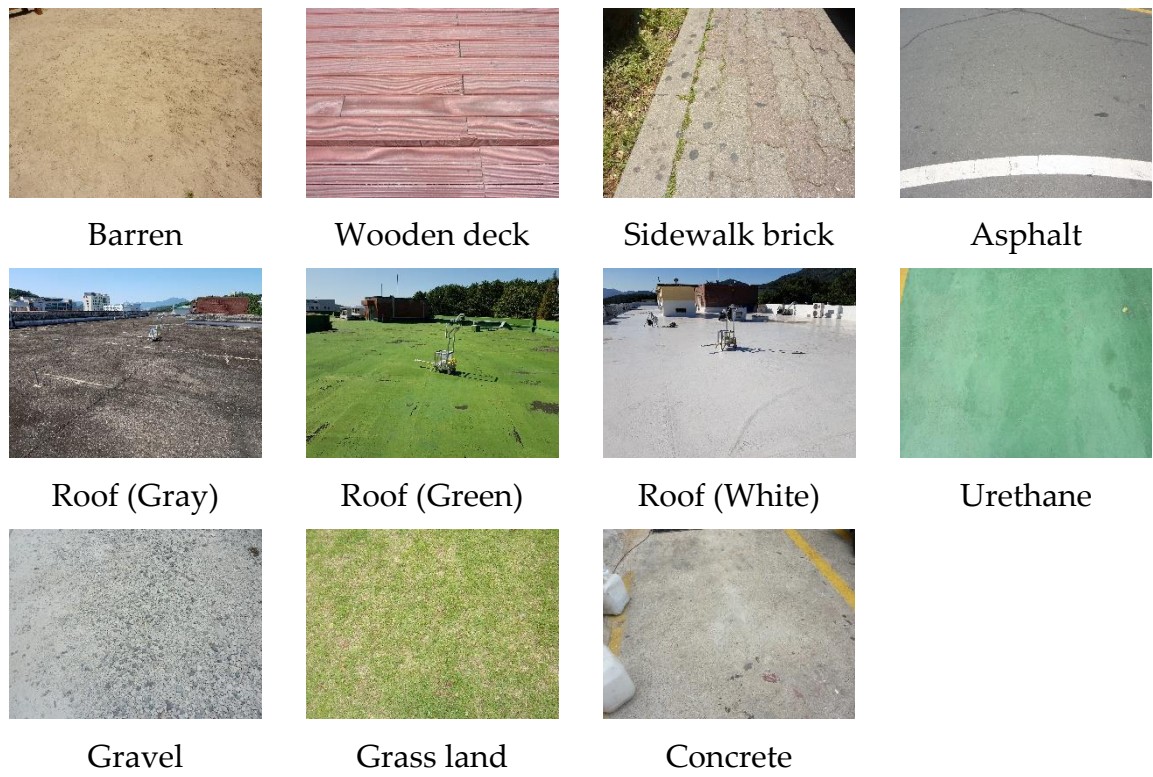

**Figure 3.** Actual images of land cover materials.

*2.4. Comparison between the UAV TIR LSTs and in-situ LSTs*

The UAV TIR LSTs were compared with the in-situ LSTs for the measurement points to verify their accuracy. For the comparison, the UAV TIR LSTs were constructed at the measurement points and their differences from the measured values were analyzed, and statistical techniques, such as scatter plot analysis, linear regression analysis, and root mean square error (RMSE) were applied. In addition, buffers with 20 and 50 cm radii were set around each measurement point to allow for the inaccuracy in the measurement points in the UAV images. The average UAV TIR LST was calculated for each buffer, and the RMSE with respect to the measured values was analyzed.

## 3. Results

*3.1. UAV TIR LST*

Figure 4 and Figure 5 showing the UAV TIR LST images. The overall LST was higher on August 2nd than on July 31st, 2019. When the LST profiles were analyzed for line A–B, the average LSTs were 40.45 °C on July 31st and 49.33 °C on August 2nd, resulting in an 8.88 °C difference. This large difference, despite only a 3-day difference in the measurements, appears to be due to a difference in solar radiation, as it was cloudy on July 31st. The nearby weather station also showed that the temperature was approximately 5 °C higher on August 2nd than on July 31st, due to the differences in solar radiation (July 31st: 32.6 °C, August 2nd: 37.2 °C).

When the LST characteristics of each land cover type were investigated through the profile analysis, it was found that the cover types with the highest LSTs were urethane (July 31st: 54.69 °C, August 2nd: 68.37 °C) and wooden deck (July 31st: 53.67 °C, August 2nd: 70.79 °C). On the other hand, the land cover type with the lowest LSTs was vegetation (July 31st: 34.30 °C, August 2nd: 38.43 °C). The LSTs of the white roof were also low (July 31st: 36.84 °C, August 2nd: 41.85 °C) and were not much different from those of the vegetation type.

When UAV TIR LST characteristics were analyzed for each measurement point, the difference in LST between the measurement dates was found to be 12.118 °C (July 31st: 45.255 °C, August 2nd: 57.373 °C), which was larger than the profile analysis result. The land cover types with the highest

LSTs were wooden deck (July 31st: 53.281 °C, August 2nd: 71.385 °C) and urethane (July 31st: 55.074 °C, August 2nd: 69.004 °C). On the other hand, vegetation (July 31st: 37.222 °C, August 2nd: 46.104 °C) and white urethane roof (July 31st: 36.313 °C, August 2nd: 41.102 °C) exhibited the lowest LSTs.

When the LST characteristics of each land cover type were analyzed from the profile analysis and measurement points, similar tendencies were observed. In particular, it was found that the LST of the white roof was even lower than that of vegetation.

In locations where the UHI frequently occurs, projects to increase the reflectivity of building roofs and create cool roofs have been actively undertaken worldwide as measures to reduce indoor and outdoor temperatures. The white urethane roof in the study area was applied to create a cool roof, and its LSTs were lower than those of concrete roofs and vegetation. The effects of cool roofs have been proven in many previous studies [30–34]. The TIR LST data from the UAV also demonstrated the temperature reduction effect of the cool roof through a comparison with other land cover types.

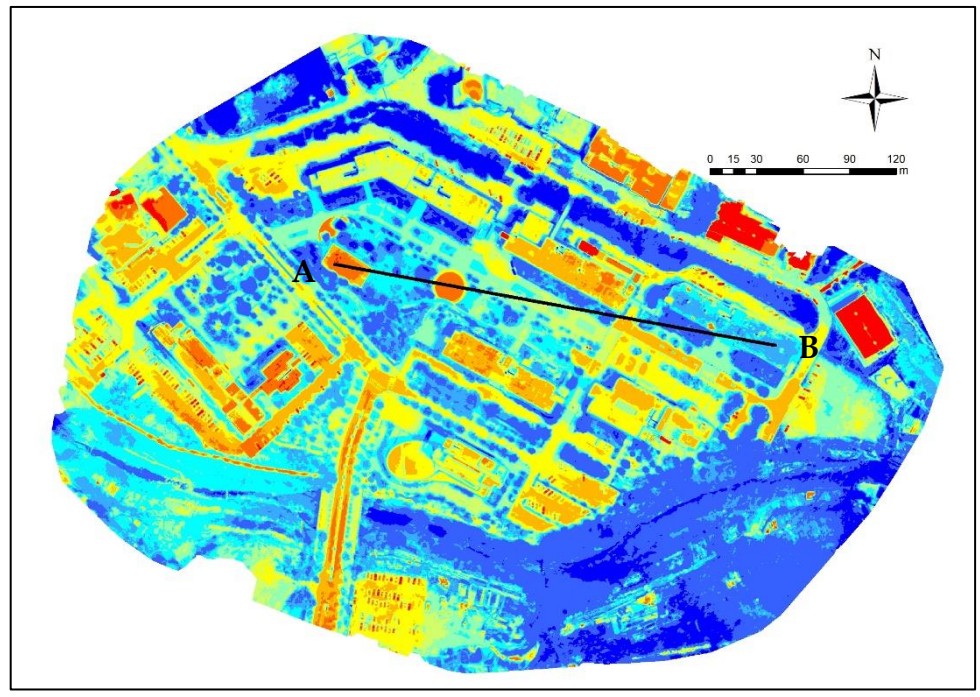

(a) 31st June 2019

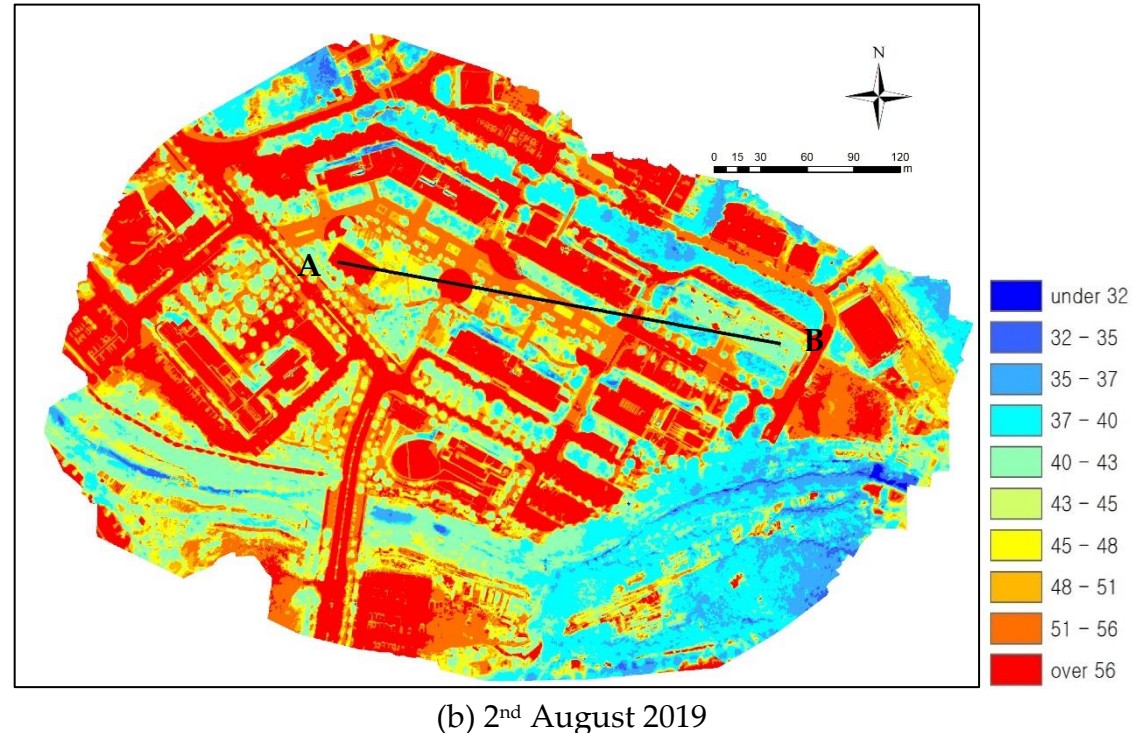

(b) 2nd August 2019

**Figure 4.** LST distribution in UAV TIR images showing the location of the A–B profile with a black line: (a) 31st June, 2019; (b) 2nd August, 2019.

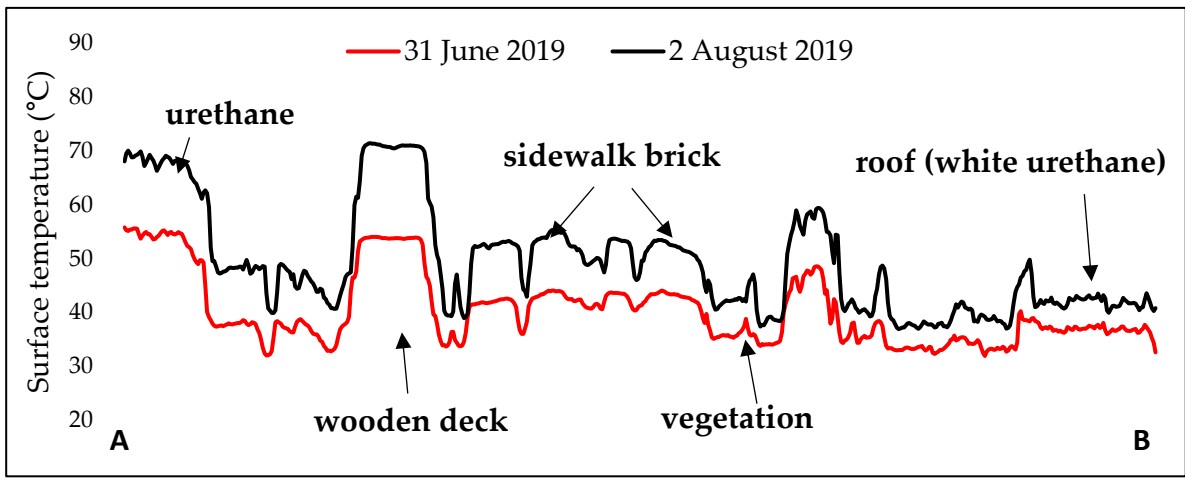

**Figure 5.** Profile analysis results for line A–B.

### 3.2. In-situ LSTs

Table 2 shows the results of measuring the LSTs of each land cover type in the study area. The land cover types with high LSTs were found to be urethane (July 31st: 61.850 °C, August 2nd: 66.667 °C) and wooden deck (July 31st: 59.939 °C, August 2nd: 70.728 °C). On the other hand, the land cover types with low LSTs were vegetation (July 31st: 37.274 °C, August 2nd: 39.677 °C) and white urethane roof (July 31st: 35.713 °C, August 2nd: 39.467 °C). The average LST for August 2nd was 7.255 °C higher than that for July 31st. The land cover types that exhibited large differences between the two dates were wooden deck (10.789 °C), asphalt (10.380 °C), and green urethane roof (10.030 °C). On the other hand, those that exhibited small differences were concrete (1.475 °C), vegetation (2.403 °C), and white urethane roof (3.754 °C).

Even the same land cover types exhibited different LSTs at different measurement points. The average standard deviation was larger on July 31st than on August 2nd (July 31st: 2.462 ℃, August 2nd: 1.93 ℃). Grassland was the land cover type with the largest standard deviation as its standard deviation was 4.555 ℃ on July 31st and 3.887 ℃ on August 2nd. Asphalt also exhibited relatively large standard deviations (July 31st: 3.110 ℃, August 2nd: 2.831 ℃). Wooden deck exhibited a large standard deviation of 3.028 ℃ on July 31st but a small value of 0.736 ℃ on August 2nd. The land cover types with small standard deviations were found to be concrete (July 31st: 1.272 ℃, August 2nd: 0.966 ℃), green urethane roof (July 31st: 1.826 ℃, August 2nd: 1.290 ℃), and white urethane roof (July 31st: 1.186 ℃, August 2nd: 0.648 ℃). It appears that even the same land cover types exhibited different LSTs because they were covered with foreign materials or they deteriorated with aging. The land cover types located on building roofs exhibited smaller standard deviations than those located on the ground surface. This is because building roofs are less likely to deteriorate without frequent human access, but the ground surface deteriorated in many cases because it mostly comprises pedestrian paths and roads. For vegetation, on the other hand, it appears that the LSTs were significantly different depending on the measurement points, not because of deterioration due to foreign materials or aging, but because vegetation and non-vegetation areas were mixed or the absorption of solar radiation varied depending on the leaf direction. For these reasons, it is necessary to reduce differences by continually securing in-situ measurement data.

### 3.3. Verification of the accuracy of the UAV TIR LSTs

#### 3.3.1. Difference between the UAV TIR LST and in-situ LST

Based on the results in Table 2, the difference between the UAV TIR LST and in-situ LST was analyzed. On July 31st, the difference was 2.672 ℃, indicating that the in-situ measurement was higher than the UAV TIR LST. On August 2nd, however, the difference was -2.191 ℃, indicating that the UAV TIR LST was higher than the in-situ LST. On July 31st, solar radiation was not consistently incident on the land surface compared to August 2nd because there were many clouds and the temperature was somewhat lower. On the other hand, on August 2nd, solar radiation was consistently incident because there were few clouds and the temperature was high. These results show that UAV TIR LST overestimated LST. As this study conducted analyses only on two dates in summer, it is necessary to compare the UAV TIR LSTs with in-situ LSTs considering seasonal factors and the influx of solar radiation.

When the land cover types were compared, wooden deck (6.658 ℃) and urethane (6.776 ℃) exhibited large differences on July 31st. They were the land cover types with the highest in-situ LSTs. On the other hand, vegetation and white urethane roof, for which the in-situ LSTs were lowest, exhibited small differences of 0.052 ℃ and -0.600 ℃, respectively. These results indicate that land cover types with higher LSTs exhibit larger differences between the UAV TIR LSTs and in-situ LSTs. On August 2nd, vegetation exhibited a large difference of -6.427 ℃ and the in-situ LST was lower than the UAV TIR LST. The other land cover types, however, showed differences of less than -2 ℃.

When such points were examined in Figure 6, the points with higher in-situ LSTs (red, over 10 ℃) and those with higher UAV TIR LSTs (black, under 10 ℃) were generally located in the vicinity of vegetation and buildings. This indicates that LSTs were not accurately detected from UAV TIR images due to the influence of shadows formed by vegetation and buildings. On August 2nd, the UAV TIR LST was found to be more than 10 ℃ higher on asphalt that was not adjacent to buildings or trees. This appears to have been because foreign materials were measured instead of asphalt. On the other hand, the difference in LST was small (less than 3 ℃) for building roofs without the influence of nearby buildings or trees. As the field of view of the TIR camera used in this study was 44° × 33°, the LST at the edge of an image could be concealed by buildings or trees. Song and Park (2014) also indicated that there were differences from the measured values in spaces with dense buildings and trees due to the off-nadir viewing angle of satellite images. Therefore, it appears that there were slight differences between the UAV TIR LSTs and in-situ LSTs when three-dimensional physical features were closely adjacent, such as buildings and trees, as a function of the field of view of the TIR camera.

**Table 2.** UAV TIR LSTs and in-situ LSTs by land cover type.

| Land cover | N | UAV | | | | In-situ | | | | Difference | |
|---|---|---|---|---|---|---|---|---|---|---|---|
| | | 31st June | | 2nd August | | 31st June | | 2nd August | | 31st | 2nd |
| | | Mean | S.D. | Mean | S.D. | Mean | S.D. | Mean | S.D. | June | August |
| Vegetation | 44 | 37.222 | 3.352 | 46.104 | 4.844 | 37.274 | 4.555 | 39.677 | 3.887 | 0.052 | -6.427 |
| Barren | 2 | 40.095 | 0.488 | 52.523 | 2.003 | 41.950 | 2.617 | 50.400 | 1.167 | 1.855 | -2.123 |
| Wooden deck | 6 | 53.281 | 0.990 | 71.385 | 0.583 | 59.939 | 3.028 | 70.728 | 0.736 | 6.658 | -0.657 |
| Sidewalk brick | 34 | 42.856 | 2.398 | 54.617 | 2.567 | 43.887 | 2.511 | 53.306 | 2.230 | 1.031 | -1.311 |
| Asphalt | 34 | 47.405 | 2.526 | 60.018 | 3.940 | 47.767 | 3.110 | 58.147 | 2.831 | 0.362 | -1.871 |
| Gravel | 8 | 44.784 | 3.001 | 56.456 | 2.316 | 44.988 | 2.074 | 54.542 | 2.612 | 0.204 | -1.914 |
| Concrete | 4 | 44.552 | 0.954 | 58.720 | 0.575 | 50.550 | 1.272 | 57.025 | 0.966 | 5.998 | -1.695 |
| Roof (Green) | 9 | 46.325 | 1.770 | 58.774 | 0.723 | 47.407 | 1.826 | 57.437 | 1.290 | 1.082 | -1.337 |
| Roof (Gray) | 10 | 49.893 | 1.499 | 62.400 | 2.205 | 50.873 | 2.496 | 59.607 | 1.968 | 0.980 | -2.793 |
| Roof (White) | 5 | 36.313 | 0.652 | 41.102 | 0.808 | 35.713 | 1.186 | 39.467 | 0.648 | -0.600 | -1.635 |
| Urethane | 4 | 55.074 | 0.626 | 69.004 | 0.484 | 61.850 | 2.409 | 66.667 | 2.908 | 6.776 | -2.337 |
| Mean | - | 45.255 | 1.660 | 57.373 | 1.913 | 47.927 | 2.462 | 55.182 | 1.931 | 2.672 | -2.191 |

S.D.: Standard deviation, N: Number

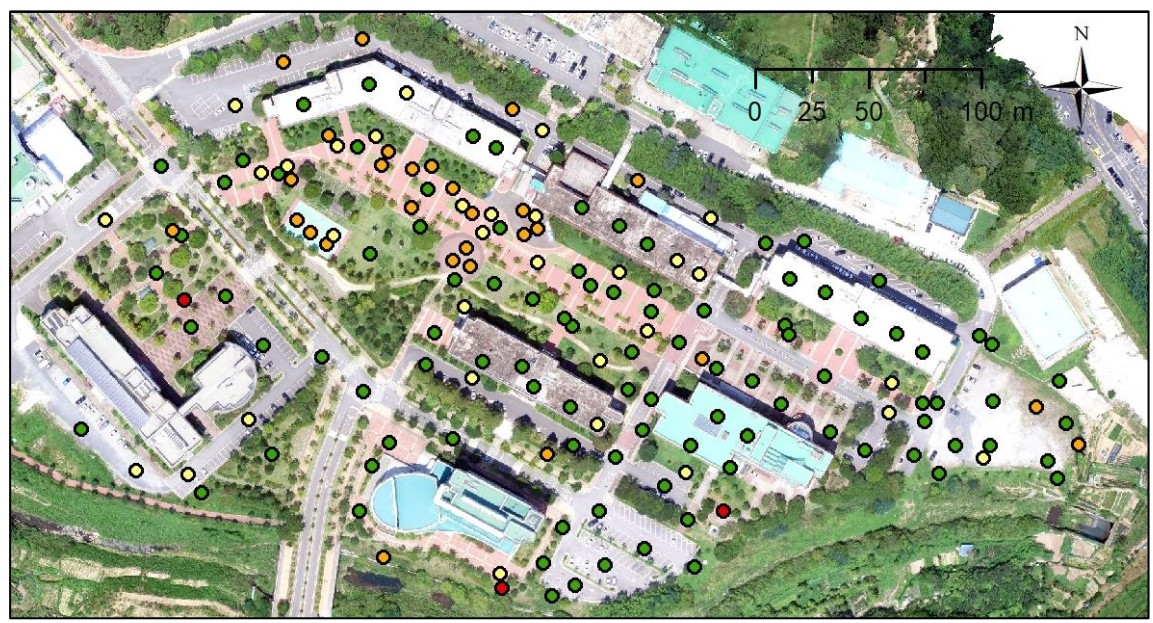

(a) 31st June 2019

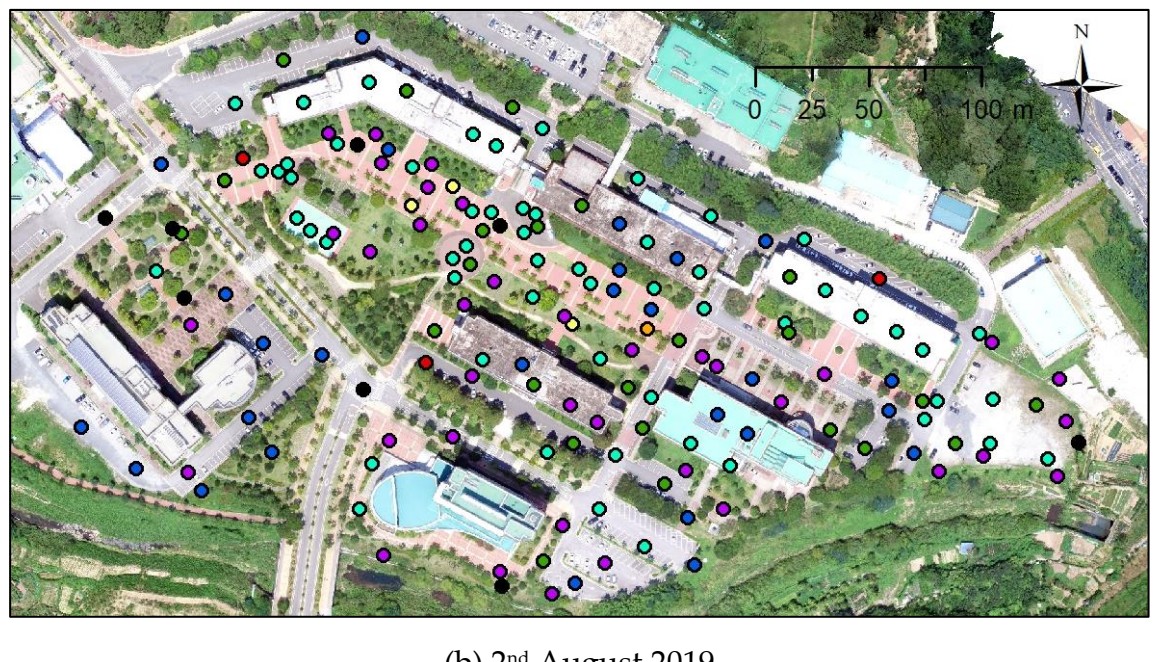

(b) 2nd August 2019

| | | | |
|---|---|---|---|
| ● **under -10°C** | ● **-5 - -3°C** | ● **0 - 3°C** | ● **5 - 10°C** |
| ● **-10 - -5°C** | ● **-3 - 0°C** | ○ **3 - 5°C** | ● **over 10°C** |

**Figure 6.** Difference between in-situ LST and UAV TIR LST at each measurement point.

### 3.3.2. Linear regression and RMSE

Figure 7 shows the results of the linear regression analysis for the in-situ LST and UAV TIR LST. Both July 31st and August 2nd exhibited very high correlations as the values of $R^2$ were higher than 0.7 (July 31st: 0.7004, August 2nd: 0.8136). In addition, the slopes were 0.6727 on July 31st and 0.7743 on August 2nd, indicating that the increment in the UAV TIR LST was smaller than that in the in-situ LST.

In Table 3, the RMSE between the in-situ LST and UAV TIR LST was analyzed. The mean RMSE was 4.030 °C on July 31st and 5.446 °C on August 2nd, indicating that the difference between the UAV TIR LST and in-situ LST was larger on July 31st. On July 31st, the RMSE values for wooden deck (7.150 °C), concrete (6.149 °C), and urethane (7.231 °C) were higher than 6 °C, and they were larger than those of the other land cover types. The standard deviation of wooden deck was relatively high (3.03 °C) in the in-situ measurement results above (Table 2), but it was smaller than the difference between the UAV TIR LST and in-situ LST. This indicates that the UAV TIR LST of wooden deck was somewhat different from the in-situ LST. For concrete and urethane, the standard deviations of the in-situ measurement results were 1.27 °C and 2.41 °C, respectively, which were smaller than the RMSE values, as was the case with wooden deck. The land cover types with small RMSE values were green urethane roof (1.973 °C) and white urethane roof (1.748 °C). They also exhibited the lowest standard deviations for the in-situ LSTs (1.83 °C and 1.19 °C, respectively).

On August 2nd, vegetation exhibited the highest mean RMSE of 8.216 °C. The standard deviation of the in-situ LSTs was 2.93 °C, which was smaller than the RMSE value, indicating that the analysis results were reliable. Vegetation also exhibited a relatively high RMSE value of 4.748 °C on July 31st. This is because the surface of leaves was measured in in-situ measurement, but some soil material could be measured instead of leaves in UAV TIR images. In addition, trees in vegetation could be affected by shadows due to their three-dimensional geometry unlike other flat land cover types, such as asphalt and concrete. For these reasons, the vegetation type appears to have exhibited large differences between the in-situ LSTs and UAV TIR LSTs. Asphalt (4.974 °C) and concrete (4.218 °C) also exhibited high RMSE values (more than 4 °C). On the other hand, white urethane roof (1.922 °C)

and green urethane roof (2.123 ℃) exhibited the lowest RMSE values, as on July 31st. Wooden deck (1.222 ℃) and concrete (2.103 ℃) showed very high RMSE values on July 31st but low RMSE values on August 2nd, thereby exhibiting the largest differences between the two dates (wooden deck: 5.928 ℃, concrete: 4.042 ℃). The land cover types that showed small differences in RMSE between the two dates were barren (0.479 ℃), sidewalk brick (0.668 ℃), gravel (0.002 ℃), green urethane roof (0.151 ℃), and white urethane roof (0.174 ℃).

When the RMSE values were compared considering buffer ranges around the measurement points, the RMSE values were largest when the buffer range was 20 cm (July 31st: 4.043 ℃, August 2nd: 5.456 ℃), but the differences with the RMSE values of the other buffer ranges were small (less than 0.04 ℃). When the differences of 20 and 50 cm buffers with the 0 cm buffer were compared by land cover type, concrete (20 cm buffer: -0.104 ℃, 50 cm buffer: -0.129 ℃) and urethane (20 cm buffer: -0.106 ℃, 50 cm buffer: 0.201 ℃) exhibited large differences in RMSE on July 31st. On August 2nd, concrete also exhibited a difference of -0.148 ℃ with a 20 cm buffer, while gray urethane roof showed a large difference of -0.278 ℃ with a 50 cm buffer. For some land cover types, the RMSE between the in-situ LST and UAV TIR LST was large or small depending on the buffer range. As the differences in RMSE were less than 0.2 ℃, however, the target points in the UAV TIR images appear to coincide with the measurement points.

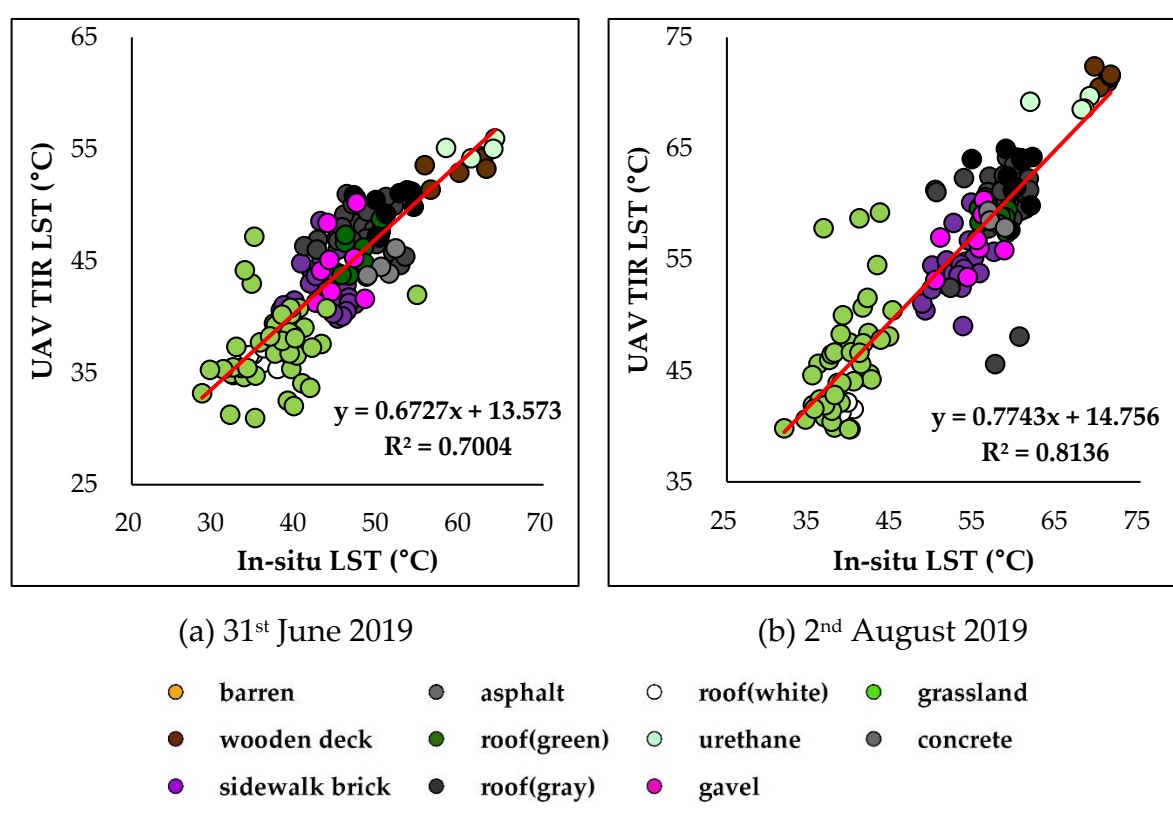

**Figure 7.** Results of the scatter plot analysis of the in-situ LST and UAV TIR LST.

**Table 3.** RMSE between the in-situ LSTs and UAV TIR LSTs (°C).

| Land cover | RMSE (°C) | | | | | | | |
|---|---|---|---|---|---|---|---|---|
| | Buffer 0 m | | Buffer 0.2 m | | Buffer 0.5 m | | Mean | |
| | 31st June | 2nd August | 31st June | 2nd August | 31st June | 2nd August | 31st June | 2nd August |
| Vegetation | 4.711 | 8.269 | 4.781 | 8.206 | 4.752 | 8.173 | 4.748 | 8.216 |
| Barren | 2.823 | 2.282 | 2.830 | 2.387 | 2.839 | 2.386 | 2.831 | 2.352 |
| Wooden deck | 7.136 | 1.232 | 7.176 | 1.229 | 7.139 | 1.206 | 7.150 | 1.222 |
| Sidewalk brick | 3.243 | 3.903 | 3.242 | 3.915 | 3.228 | 3.900 | 3.238 | 3.906 |
| Asphalt | 3.235 | 4.957 | 3.224 | 5.016 | 3.220 | 4.949 | 3.226 | 4.974 |
| Gravel | 3.259 | 3.275 | 3.195 | 3.242 | 3.273 | 3.205 | 3.242 | 3.241 |
| Concrete | 6.063 | 2.014 | 6.167 | 2.162 | 6.216 | 2.143 | 6.149 | 2.106 |
| Roof (Green) | 1.998 | 2.117 | 1.980 | 2.154 | 1.940 | 2.099 | 1.973 | 2.123 |
| Roof (Gray) | 2.817 | 4.235 | 2.387 | 4.252 | 2.791 | 4.168 | 2.665 | 4.218 |
| Roof (White) | 1.766 | 1.928 | 1.712 | 1.874 | 1.767 | 1.965 | 1.748 | 1.922 |
| Urethane | 7.140 | 3.805 | 7.246 | 3.870 | 7.307 | 3.604 | 7.231 | 3.760 |
| Total | 4.012 | 5.466 | 4.043 | 5.456 | 4.034 | 5.416 | 4.030 | 5.446 |

## 4. Discussion

In this study, UAV TIR LSTs were compared with in-situ LSTs measured at specific measurement points, and the accuracy of the UAV TIR LSTs was examined through linear regression and RMSE analysis. The difference between the UAV TIR LST and in-situ LST varied depending on the land cover type, and it was also affected by physical factors, such as nearby vegetation and buildings. Moreover, it was found that the accuracy of the UAV TIR LSTs was affected by weather conditions, such as the influx of solar radiation and clouds. As a result of these various factors, the RMSE between the UAV TIR LSTs and in-situ LSTs was found to vary from 4 to 5 °C. The land cover types with high LSTs, such as wooden deck, urethane, and concrete, generally exhibited higher RMSE values. As shown in the results of linear regression analysis, the coefficient of determination ($R^2$) of the model was higher than 0.7, indicating a very high correlation.

As for previous studies that compared LST data acquired using satellite images or UAVs with in-situ LSTs, Kraaijenbrink et al. (2018) measured and compared UAV TIR LSTs, Landsat 8 images, and in-situ LSTs of areas covered with glaciers [19]. The UAV TIR LSTs exhibited differences of -1.4 ± 1.8, 11.0 ± 5.2, and 15.3 ± 4.7 °C in three flights. Kelly et al. (2019) analyzed the accuracy of a non-radiometric FLIR Vue Pro 640 camera mounted on UAVs based on laboratory and field experiments [22]. While the accuracy was stable (approximately 0.5 °C) under laboratory conditions, it decreased to 5 °C in the field experiments due to the ambient conditions. They pointed to the non-linear relationship between the camera output and the sensor temperature under the influence of the wind and temperature generated during the UAV flight as the cause of performance degradation. As for studies on TIR satellite images, Song and Park (2014) reported that the difference between satellite image LSTs and in-situ LSTs varied depending on time, and that the difference could be larger than 10 °C in summer when the temperature is high [24]. Voogt and Oke (2003), Hartz et al. (2006), Barring et al. (1985), and Eliasson (1992) reported that the difference between satellite image LSTs and in-situ LSTs was large in areas with very dense buildings due to the limited horizontal surface view [11,25–27]. According to these authors, this phenomenon is caused by the accumulation of Earth radiation energy released to the atmosphere, which is large in spaces with dense buildings [35], and this cannot be detected by satellite images.

These results indicate that the LST data captured from satellite images and UAVs are different from in-situ LSTs for various reasons. The UAV TIR LSTs acquired in this study also appear to be different from the in-situ LSTs due to several causes, such as weather conditions in the atmosphere, the field of view of the camera, and the camera certification, as in previous studies. Also, it is

determined that an error in measurement points may cause problems in the accuracy of the UAV TIR LSTs. These causes need to be clearly identified through the results of systematic experimentation. Furthermore, in order to alleviate the UHI, it is important to accurately identify the thermal characteristics of various land cover materials and spatial factors present in urban areas. This study revealed that the accuracy of the UAV TIR LSTs varies depending on the land cover material. This is a meaningful result in terms of the utilization of UAV TIR images in studying the alleviation of the UHI and indicates further research is required to improve accuracy.

## 5. Conclusions

In this study, the accuracy of the LSTs acquired from a UAV TIR camera was verified in a university campus area featuring various land cover materials. To this end, the UAV TIR LSTs were compared with in-situ LSTs for 160 measurement points on two dates (July 31st and August 2nd).

Both the UAV TIR LSTs and in-situ LSTs were high for asphalt, wooden deck, and urethane, but they were low for trees and lawns. The LSTs for the white urethane roof, referred to as a cool roof, were lower than those of trees and lawns. When the UAV TIR LSTs were compared with in-situ LSTs, the latter were 2.672 °C higher than the UAV TIR LSTs on July 31st, when the temperature was low, but the UAV TIR LSTs were 2.191 °C higher on August 2nd, when the temperature was high. The results of linear regression analysis show $R^2$ values higher than 0.7, indicating a high correlation between the UAV TIR LSTs and in-situ LSTs. The RMSE values were 4.030 °C on July 31st and 5.446 °C on August 2nd, and the RMSE values varied depending on the land cover material. These results show that the UAV TIR LSTs were somewhat different from the in-situ LSTs. Various factors, such as weather conditions, UAV operation, and the certification and field of view of the TIR camera, appear to have caused these differences, which had been pointed out in previous studies.

In order to effectively alleviate the UHI, it is extremely important to identify the thermal characteristics of various spatial factors distributed in urban areas. In this respect, UAVs can be used effectively to identify the LST and thermal characteristics of each land cover material. The accuracy of the UAV TIR LST data, however, needs to be verified to accurately diagnose the UHI and to implement measures for its alleviation. In this study, there were problems with the accuracy of UAV TIR LST data derived for various land cover materials in urban areas. The causes of these problems need to be identified through further research. Based on this, the utilization of UAVs should be gradually expanded to help alleviate the UHI and to mitigate heat waves and improve thermal comfort in urban areas.

**Author Contributions:** B.S. and K.P. conceived and designed the research; B.S. performed the field measurements; and K.P. and B.S. analyzed the spatial data and wrote the paper. All authors have read and agreed to the published version of the manuscript.

**Funding:** This research was supported by the Basic Science Research Program through the National Research Foundation of Korea (NRF) funded by the Ministry of Education (No. NRF-2019R1I1A1A01063568).

**Conflicts of Interest:** The authors declare no conflicts of interest.

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
