# Peer review of "Verification of Accuracy of Unmanned Aerial Vehicle (UAV) Land Surface Temperature Images Using In-Situ Data"

_remotesensing, doi:10.3390/rs12020288_

Round 1

Reviewer 1 Report

line 16: Must be rounded to 4, it is an error.

line 17: Rounded to 5

line 52-54: It would be also fair to list the disadvantages of UAV applications.

line 66-68: I encourage the reading and citation of Garcia-Santos et al. (2019): Study of Temperature Heterogeneities at Sub-Kilometric Scales and Influence on Surface–Atmosphere Energy Interactions.

line 110: It is mandatory to explain how LST was estimated from FLIR UAV TIR camera. Atmospheric correction between surface and UAV height, LST algorithm used, etc.

line 111: The Camera has an emissivity? that is odd.

line 130: I understood this is the emissivity set up, presumed to be the corresponding to the measured surface, right?, So why TIR camera has the emissivity set to 0.98? maybe this is inducing an error to the validation proccess.

line 133-134: Why is not explained the GPS recording of the LST ground measurements? Maybe it was stipulated with some marks the places to be measured? anyway, it must be explined in the text.

line 149: Do you mean the average vaue?

line 184-192: I think the paragraph is more or less repeated from above section.

line 237-240: An also because there is a problem with the GPS recording of the measured data. You don't know what spot are you comparing with the UAV TIR data.

Author Response

Point 1: line 16: Must be rounded to 4, it is an error.

Response 1: We have modified line 16.

Point 2: line 17: Rounded to 5

Response 2: We did not understand your comment (line 17: rounded to 5). Please let me know your comment.

Point 3: line 52-54: It would be also fair to list the disadvantages of UAV applications.

Response 3: The paragraph that the reviewer points out is a list of the advantage of UAV, and we don’t think it is appropriate to mention the disadvantages of UAV applications.

 Point 4: line 66-68: I encourage the reading and citation of Garcia-Santos et al. (2019): Study of Temperature Heterogeneities at Sub-Kilometric Scales and Influence on Surface–Atmosphere Energy Interactions.

Response 4: I think the papers recommended by the reviewers are very meaningful. This paper is also limited to buildings, roads and grass to analyse the accuracy of UAV LST. Therefore, it is considered to be suitable as a reference for the necessity of verifying the accuracy of UAV LST for various land cover materials which is the purpose of this study.

Point 5: line 110: It is mandatory to explain how LST was estimated from FLIR UAV TIR camera. Atmospheric correction between surface and UAV height, LST algorithm used, etc.

Response 5: FLIR UAV TIR camera does not perform atmospheric correction based on UAV height. The equation for calculating the LST from the FLIR UAV TIR camera is presented in the paper.

Point 6: line 111: The Camera has an emissivity? that is odd.

Response 6: FLIR Vue Pro R camera has an emissivity of 0.98.

Point 7: line 130: I understood this is the emissivity set up, presumed to be the corresponding to the measured surface, right?, So why TIR camera has the emissivity set to 0.98? maybe this is inducing an error to the validation proccess.

Response 7: We set the emissivity of FLIR camera and infrared thermometer equally to 0.98. The emissivity value in line 130 is error. So we modified to 0.98.

Point 8: line 133-134: Why is not explained the GPS recording of the LST ground measurements? Maybe it was stipulated with some marks the places to be measured? anyway, it must be explined in the text.

Response 8: Field measurements were made to select a point to measure in advance so that the investigator could make a measurement at that point. The reason is that it takes a lot of time to record the measurement and location information at the same time. Of course, it is important to measure the surface temperature at the correct position. We made three measurements around the measurement point to minimize errors at the measurement point. It is judged that the error of surface temperature would not be large because the same land covering material was around the measuring points.

Point 9: line 149: Do you mean the average vaue?

Response 9: The average LST means the average value of the surface temperature corresponding to the profile line.

Point 10: line 184-192: I think the paragraph is more or less repeated from above section.

Response 10: This section describes the surface temperature characteristics for field measurements and differs from the UAV LST results above. It can be thought of as repeating as reviewer pointed out, showing similar patterns depending on land cover material and timing. However, We think it is necessary to separate the two sections and present their characteristics.

Point 11: line 237-240: An also because there is a problem with the GPS recording of the measured data. You don't know what spot are you comparing with the UAV TIR data.

Response 11: We think there is also a problem with the GPS record of the measurement point that the reviewer pointed out. This is considered to be the limitation of this study and it should be improved in the future. This problem is presented in the Discussion section. 

We corrected and supplemented by the reviewer’s points. Thank you for your careful review of the paper.

Reviewer 2 Report

UAV can help to acquire very high-resolution LST images and investigate urban climate. The topic of the study is interesting and in the scope of the journal. The author verified the UAV LST data using a large number of in-situ observation data. The results can provide implication for the future application of UAV technique to study UHI. However, some key information about the LST data inversion is not presented. The structure of results analysis needs adjustment. Some statements need further clarification. Given all these concerns, I suggest a major revision for further consideration.

Title

UAV: it is better to show the full name of UAV.

Abstract

(1) It is better to show the full name of TIR when showing it for the first time.

(2) “Accurately diagnosing the thermal characteristics of urban areas based on the spatial elements is necessary to address the heat island phenomenon.” It seems to be better to show the motivation at the beginning of the abstract. Consider it.

Introduction

Urban heat island is a very common definition, you can use abbreviation format UHI after the first description of the full name.

Line 28-30: the description of UHI mechanism is not insufficient. In the daytime, it is mainly caused by the difference in evapotranspiration and radiation absorption and at night it is mainly caused by the difference in the heat storage of different surface.

 Line 32: what is the meaning of various spatial characteristics? Do you mean different land cover or imperviousness?

https://www.sciencedirect.com/science/article/pii/S0048969718338993

Line 35-47: an important advantage of the satellite is it can be used to monitor UHI at a large scale, from a city to global. You can add the related description.

It is better to strengthen the innovation of the study. For example, a large number of in-situ observations on different land cover.

Method

(1) Please add the description of the algorithm to calculate the LST using TRI images from UAV and in-situ observation and the values of the key parameters, such as emissivity.

(2) Please add more descriptions about the technique flow of LST inversion from TRI images.

(3) Line 113-114: you did three times UAV operation. Did you use all the images when calculating LST and how did you use them to calculate the final LST image?

(3) Line 116: you mentioned the resolution was set 0.2cm. But in line 78, you mentioned LST images of 1cm resolution was collected. Why two different resolution here?

(4) Line 133-134: how do you eliminate the bias caused by the different measurement time when you compare the measurement at different points and compare with the UAV data? Did you do data correction?

Result

Figure 5. Please add the results of in-situ observation along with the profile in this figure.

Line 193: Is the difference caused by the different measurement time?

Lin3 220-221: How do you conclude that “These results show that UAV TIR LST 220 overestimated LST.”

Figure 6: Please add a legend about the different color lines and the equations. It is not necessary to show R 2 and R together in the same figure.

Line 231-250: Does figure 6 include the data on both two days? If it is, the figure includes information on both spatial and temporal variation. The positive bias from July 31 and negative bias from 2, august. I suggest drawing the figure for the two days separately.

Line 242-243: This appears to have been because foreign materials were measured instead of asphalt. Not clear.

The structure of 3.3 need adjustment. The RMSE is also an indicator of two data difference. For example, the analysis of table 3 should be moved to section 3.3.1.

Discussion

Line 331-332: not clear.

Author Response

Point 1: UAV: it is better to show the full name of UAV.

Response 1: We have revised your comment in this paper.

Abstract

Point 2: (1) It is better to show the full name of TIR when showing it for the first time.

Response 2: We have revised your comment in this paper.

Point 3: (2) “Accurately diagnosing the thermal characteristics of urban areas based on the spatial elements is necessary to address the heat island phenomenon.” It seems to be better to show the motivation at the beginning of the abstract. Consider it.

Response 3: Good point. We corrected the sentence as the reviewer pointed out.

Introduction

Point 4: Urban heat island is a very common definition, you can use abbreviation format UHI after the first description of the full name.

Response 4: We changed the urban heat island to UHI.

Point 5: Line 28-30: the description of UHI mechanism is not insufficient. In the daytime, it is mainly caused by the difference in evapotranspiration and radiation absorption and at night it is mainly caused by the difference in the heat storage of different surface.

Response 5: We have revised your comment in this paper.

Point 6: Line 32: what is the meaning of various spatial characteristics? Do you mean different land cover or imperviousness?

Response 6: Various spatial characteristics mean various land cover types.

Point 7: Line 35-47: an important advantage of the satellite is it can be used to monitor UHI at a large scale, from a city to global. You can add the related description.

Response 7: We have revised your comment in this paper.

Point 8: It is better to strengthen the innovation of the study. For example, a large number of in-situ observations on different land cover.

Response 8: The advantage of this study is that field measurements were taken into consideration for many points on various land cover materials. This part was modified to highlight.

Method

Point 9: (1) Please add the description of the algorithm to calculate the LST using TRI images from UAV and in-situ observation and the values of the key parameters, such as emissivity.

Response 9: An equation for calculating LST extracted from UAV FLIR Vue Pro R TIR camera and infrared thermometer is presented. Longwave radiation detected by the FLIR TIR camera measures the radiant energy by fixing the emissivity at 0.98. Therefore, the same conditions were considered by setting the emissivity to 0.98 in the field measurement. The sentence was revised to convey the contents well.

Point 10: (2) Please add more descriptions about the technique flow of LST inversion from TRI images.

Response 10: We have revised your comment in this paper.

Point 11: (3) Line 113-114: you did three times UAV operation. Did you use all the images when calculating LST and how did you use them to calculate the final LST image?

Response 11: We captured UAV LST images once on July 31st and August 2nd.  

Point 12: (3) Line 116: you mentioned the resolution was set 0.2cm. But in line 78, you mentioned LST images of 1cm resolution was collected. Why two different resolution here?

Response 12: We produced UAV LST images with 2cm resolution. 1cm of the introduction was corrected to 2cm.

Point 13: (4) Line 133-134: how do you eliminate the bias caused by the different measurement time when you compare the measurement at different points and compare with the UAV data? Did you do data correction?

Response 13: We completed the measurements within 30 minutes to minimize surface temperature change. Since the surface temperature change was less than 1 degree with time, we did not perform temperature correction.

Results

Point 14: Figure 5. Please add the results of in-situ observation along with the profile in this figure.

Response 14: Figure 5 shows the UAV LST results for the profile line. The measurement points do not correspond to the profile line and the results cannot be presented in the same figure.

Point 15: Line 193: Is the difference caused by the different measurement time?

Response 15: Table 5 presents the standard deviation of the measured surface temperature for each land cover type. As a result, the surface temperature difference with respect to the measurement time can be predicted.

Point 16: Lin3 220-221: How do you conclude that “These results show that UAV TIR LST 220 overestimated LST.”

Response 16: On August 2, when there was no cloud, it was judged that there were few variables such as weather condition. At this time, because UAV LST was about 2 degrees higher than field measurement LST, it was judged that UAV LST predicted the LST excessively.

Point 17: Figure 6: Please add a legend about the different color lines and the equations. It is not necessary to show R 2 and R together in the same figure.

Response 17: We have revised your comment in this paper.

Point 18: Line 231-250: Does figure 6 include the data on both two days? If it is, the figure includes information on both spatial and temporal variation. The positive bias from July 31 and negative bias from 2, august. I suggest drawing the figure for the two days separately.

Response 18: Figure 6 shows the characteristics of the LST difference between UAV and field measurement according to UAV LST, but did not identify any particulars. The results into two days separately suggested by the reviewer also showed no meaningful results (R2 0.001 or less). We determined that this data should be deleted because it does not significantly affect the results.

Point 19: Line 242-243: This appears to have been because foreign materials were measured instead of asphalt. Not clear.

Response 19: It was carefully measured not to measure foreign materials in the field measurement.

Discussion

Point 20: Line 331-332: not clear.

Response 20: According to Erell et al.(2012), the lower sky view factor (SVF), the denser the buildings, the more the Earth’s radiant energy (long wave radiance) is accumulated because of reflected to the building. The sentence was revised to express this meaning well. 

We corrected and supplemented by the reviewer’s points. Thank you for your careful review of the paper.

Reviewer 3 Report

Verification of Accuracy of UAV Land Surface Temperature Images Using In-Situ Data

by Song and Park

General comments:

The drone has been widely used to the study related to deriving physical properties of the land surface, this is an example that illustrates how a drone can be used for deriving surface temperature application. In this work, the authors show the techniques for verifying the accuracy of UAV TIR surface temperature of various land cover by using in situ data.

This study is quite interesting; however, I have comments and suggestions that should be addressed by the authors in order to improve the manuscript:

Major comments

In your FLIR UAV, you use emissivity 0.98 for your camera and on in situ thermometers you have 0.95, Why you set difference emissivity on both types of equipment? Is it fix value, or you can set it? (as far I know we can set the FLIR emissivity, but I don’t know for the thermometer). Since emissivity is very important for deriving temperature you have to take care with this emissivity (noted difference even only 0.01 in emissivity could lead difference) see (Feijt & Kohsiek, 1995). Could you extend your discussion with considering this effect? Another comment related to the effect of flight altitude to temperature sensitivity. In your study, you had a fixed flight altitude of 150 m. Can you consider the effect of acquisition altitude to the temperature results? Since this is important due to thermal mixing phenomena. (see Ramirez-González et al., 2019, Zhang et al., 2019)

Feijt, A. J., & Kohsiek, W. (1995). The effect of emissivity variation on surface temperature determined by infrared radiometry. Boundary-Layer Meteorology, 72(3), 323–327. https://doi.org/10.1007/BF00836339

Ramirez-González, L. M., Aufaristama, M., Jónsdóttir, I., Höskuldsson, Á., Póroarson, P., Proietti, N. M., … McQuilkin, J. (2019). Remote sensing of surface Hydrothermal Alteration, identification of Minerals and Thermal anomalies at Sveifluháls-Krýsuvík high-temperature Geothermal field, SW Iceland. IOP Conference Series: Earth and Environmental Science, 254(1). https://doi.org/10.1088/1755-1315/254/1/012005

Zhang, L., Niu, Y., Zhang, H., Han, W., Li, G., Tang, J., & Peng, X. (2019). Maize Canopy Temperature Extracted From UAV Thermal and RGB Imagery and Its Application in Water Stress Monitoring. Frontiers in Plant Science, 10(October), 1–18. https://doi.org/10.3389/fpls.2019.01270

Author Response

Point 1: In your FLIR UAV, you use emissivity 0.98 for your camera and on in situ thermometers you have 0.95, Why you set difference emissivity on both types of equipment? Is it fix value, or you can set it? (as far I know we can set the FLIR emissivity, but I don’t know for the thermometer). Since emissivity is very important for deriving temperature you have to take care with this emissivity (noted difference even only 0.01 in emissivity could lead difference) see (Feijt & Kohsiek, 1995). Could you extend your discussion with considering this effect? Another comment related to the effect of flight altitude to temperature sensitivity. In your study, you had a fixed flight altitude of 150 m. Can you consider the effect of acquisition altitude to the temperature results? Since this is important due to thermal mixing phenomena. (see Ramirez-González et al., 2019, Zhang et al., 2019)

Response 1: In field measurement, the emissivity of the infrared thermometer was set to 0.98. The content of the text has been corrected with typos. And, depending on the flight altitude, the accuracy of the LST may vary depending on the weather effect in the atmosphere. This study is to be pursued in the future. In this study, it is fixed to 150m and a comparative comparison of land cover materials. 

We corrected and supplemented by the reviewer’s points. Thank you for your careful review of the paper.

Round 2

Reviewer 1 Report

Authors Response 2: We did not understand your comment (line 17: rounded to 5). Please let me know your comment.

Reviewer: It means that the error must be rounded (e.g., RMSE 5.446 °C, so it must be rounded to 5 °C). Do the same with 4.030 °C in the same line, ant the corresponding part of the text.